# Usefulness of Sepsis-3 in diagnosing and predicting mortality of ventilator-associated lower respiratory tract infections

**Alexandre Gaudet**[1,2], **Matthieu Devos**[1], **Sylvain Keignart**[1], **Olivier Pouly**[1], **Sylvain Lecailtel**[1], **Frédéric Wallet**[3], **Saad Nseir**[1,4]*

1 Department of Intensive Care Medicine, Critical Care Center, CHU Lille, Lille, France, 2 Univ. Lille, CNRS, Inserm, CHU Lille, Institut Pasteur de Lille, U1019-UMR9017-CIIL-Centre d'Infection et d'Immunité de Lille, Lille, France, 3 Microbiology Laboratory, CHU Lille, Lille, France, 4 Team Fungal Associated Invasive & Inflammatory Diseases, Lille Inflammation Research International Center, INSERM U995, Université de Lille, Lille, France

* s-nseir@chru-lille.fr

## Abstract

### Background

Early distinguishing ventilator-associated tracheobronchitis (VAT) and ventilator-associated pneumonia (VAP) remains difficult in the daily practice. However, this question appears clinically relevant, as treatments of VAT and VAP currently differ. In this study, we assessed the accuracy of sepsis criteria according to the Sepsis-3 definition in the early distinction between VAT and VAP.

### Methods

Retrospective single-center cohort, including all consecutive patients with a diagnosis of VAT (n = 70) or VAP (n = 136), during a 2-year period. Accuracy of sepsis criteria according to Sepsis-3, total SOFA and respiratory SOFA, calculated at time of microbiological sampling were assessed in differentiating VAT from VAP, and in predicting mortality on ICU discharge.

### Results

Sensitivity and specificity of sepsis criteria were found respectively at 0.4 and 0.91 to distinguish VAT from VAP, and at 0.38 and 0.75 for the prediction of mortality in VA-LRTI. A total SOFA ≥ 6 and a respiratory SOFA ≥ 3 were identified as the best cut-offs for these criteria in differentiating VAT from VAP, with sensitivity and specificity respectively found at 0.63 and 0.69 for total SOFA, and at 0.49 and 0.7 for respiratory SOFA. Additionally, for prediction of mortality, a total SOFA ≥ 7 and a respiratory SOFA = 4 were identified as the best-cut-offs, respectively yielding sensitivity and specificity at 0.56 and 0.61 for total SOFA, and at 0.22 and 0.95 for respiratory SOFA.

**Data Availability Statement:** All relevant data are within the manuscript and its Supporting information files.

**Funding:** The author(s) received no specific funding for this work.

**Competing interests:** I have read the journal's policy and the authors of this manuscript have the following competing interests: SN received fees from MSD, Pfizer, Gilead, BioMérieux, and Bio Rad for lectures. Other authors declare that they have no competing interest. This does not alter our adherence to PLOS ONE policies on sharing data and materials.

**Abbreviations:** CI, confidence interval; CT, computed tomography; ICU, intensive care unit; IQR, interquartile range; NPV, negative predictive value; PPV, positive predictive value; SD, standard deviation; Se, sensitivity; Sp, specificity; SOFA, sequential organ failure assessment; VA-LRTI, ventilator-associated lower respiratory tract infection; VAP, ventilator-associated pneumonia; VAT, ventilator-associated tracheobronchitis.

## Conclusions

Sepsis criteria according to the Sepsis-3 definition show a high specificity but a low sensitivity for the diagnosis of VAP. Our results do not support the use of these criteria for the early diagnosis of VAP in patients with VA-LRTI.

## Background

Despite decades of research, ventilator-associated lower respiratory tract infections (VA-LRTI), including ventilator-associated tracheobronchitis (VAT) and ventilator-associated pneumonia (VAP), remain frequent complications of mechanical ventilation [1]. The distinction between these two conditions currently relies on the interpretation of chest X-ray, and is based on the presence of a new pulmonary infiltrate in VAP, conversely to VAT [2].

The diagnosis of VA-LRTI is associated with poor outcomes, including longer duration of mechanical ventilation and increased length of stay in the intensive care unit (ICU) [1]. Current IDSA/ATS guidelines recommend the early initiation of antimicrobial therapy in patients with VAP, but not in those with VAT [3]. Therefore, making the early distinction between VAT and VAP appears as a clinically relevant question in mechanically ventilated patients with suspected VA-LRTI.

However, numerous studies have highlighted the difficulties of interpretation of chest radiographies in differentiating VAT from VAP, mainly due to other potential etiologies of lung opacities [4, 5] and to a frequent delay in the appearance of pulmonary infiltrates [6]. Several studies suggest that a diagnosis of VAP would be associated with a greater severity of illness, thus explaining higher mortality rates than in VAT [1, 7, 8]. Accordingly, in the TAVeM study, Martin-Loeches *et al.* reported higher values of the SOFA score on the day of diagnosis of VA-LRTI in patients with VAP, compared to those with VAT.

In 2016, the Sepsis-3 Definition Task Force developed and released new criteria for the diagnosis of sepsis, based on the assessment of the severity of illness. Based on these criteria, sepsis is defined as a change in total SOFA score $\geq 2$ points consequent to the infection over a 48 hours period [9]. Thus, patients with a diagnosis of VAP may be more likely to experience a greater severity of illness, notably characterized by a worsening in organ failures, therefore leading to a higher frequency of sepsis. Accordingly, we aimed to evaluate the accuracy of sepsis criteria according to the Sepsis-3 definition in differentiating VAT from VAP.

## Methods

### Study design and patients

This study was conducted in a 50-bed mixed ICU (Department of Intensive Care Medicine, Critical Care Centre, CHU of Lille), during a 2-year period (from January 1st, 2016 to December 31, 2017). Continuous surveillance of ICU-acquired infections allowed prospective identification of patients with VA-LRTI. These patients were subsequently included in this retrospective study and other data were extracted from electronic files.

### Ethics statement

This research was examined and validated by the Institutional Review Board of the University Hospital of Lille (CPP Nord Ouest IV) under number HP 20/37. Following IRB recommendations, in accordance with the French law, and because of the retrospective observational design, written informed consent was not required.

## Data collection

Patient demographic characteristics, severity scores, comorbidities, primary diagnoses, prior antibiotic exposure were recorded at baseline for all patients. Furthermore, data about clinical, biological, and radiological diagnostic criteria for VA-LRTI, microbiological diagnostic procedures, microbiological findings, degree of severity on the onset of infection, antibiotic use and clinical outcomes were obtained.

## Definitions

Criteria from the International ERS/ESICM/ESCMID/ALAT guidelines for the management of hospital-acquired pneumonia and ventilator-associated pneumonia were used for the definition of VAT and VAP (S1 Appendix) [2].

Accordingly, diagnosis of VA-LRTI was based on the presence of at least 2 of the following criteria: body temperature of more than 38.5˚C or less than 36.5˚C, leucocyte count greater than 12 000 cells per μL or less than 4 000 cells per μL, and purulent endotracheal aspirate. Microbiological confirmation was needed for all episodes of infection, with the isolation in the endotracheal aspirate of at least $10^5$ CFU per mL, or in bronchoalveolar lavage of at least $10^4$ CFU per mL.

VAT was defined as the association of the above-mentioned criteria with no radiographical signs of new pneumonia. Conversely, VAP was defined by the presence of new or progressive infiltrates on chest X-ray along with these criteria [2]. CT-scan images were not used to distinguish VAT from VAP. Only first episodes of VAT and VAP were taken into account for this study.

The diagnosis of sepsis was made in accordance with the Sepsis-3 criteria, and was therefore defined as a change in total SOFA score $\geq 2$ points consequent to the infection over a 48 hours period before collection of the respiratory sample used for the microbiological confirmation of VA-LRTI [9]. Accordingly, we calculated the $\Delta_{SOFA}$ as the difference between SOFA scores calculated at the time of microbiological sampling, and 48h before microbiological sampling. The diagnosis of sepsis was established in case of $\Delta_{SOFA} \geq 2$ points.

## Objectives

The primary aim of this study was to evaluate the accuracy of Sepsis-3 criteria in differentiating VAT from VAP in patients with microbiologically confirmed VA-LRTI. The secondary aims of this study were to assess the accuracy of total SOFA score and respiratory SOFA in differentiating VAT from VAP and to evaluate the accuracy of Sepsis-3 criteria, total SOFA score and respiratory SOFA in predicting mortality in patients with microbiologically confirmed VA-LRTI.

## Statistical analysis

Categorical variables were expressed as numbers (percentages) and compared using Chi-square test or Fisher's exact test, as appropriate. Normality of distribution of continuous variables was checked graphically and by using the Shapiro–Wilk test. Skewed continuous variables were presented as median (interquartile range) and compared using Mann-Whitney U test. Normally distributed continuous variables were presented as means (SD), and compared using Student's t-test.

We assessed the accuracy of $\Delta_{SOFA}$, total SOFA and SOFA respiratory at the time of microbiological sampling in differentiating VAT from VAP and in predicting mortality by calculating their sensitivity (Se), specificity (Sp), positive and negative predictive values (PPV and NPV) as well as positive and negative likelihood ratios.

All statistical tests were two-tailed, and p values <0.05 were considered statistically significant. Statistical analysis was performed using SPSS 22.0 (IBM, New York, NY) software.

## Results

### Patient characteristics

Seventy patients with VAT and 136 patients with VAP were included in this study. Study patient characteristics are shown in Table 1. Compared to VAT, patients with a diagnosis of VAP had higher SOFA score on ICU admission (mean (SD) 8.9 (4) vs 7.2 (4.3), p = 0.008) and lower percentage of chronic heart disease (12% vs 23%, p = 0.037), while there was no significant difference for clinical outcomes between the two groups (Table 1). Furthermore, we found a higher frequency of *Enterobacter* spp. (15% vs 6%, p = 0.043) and a lower frequency of *Citrobacter freundii* (1% vs 6%, p = 0.047) in patients with VAP than in those with VAT (Table 2).

### Accuracy of sepsis criteria according to Sepsis-3 in differentiating VAT from VAP, and predicting mortality

A diagnosis of sepsis was found more frequently in VAP than in VAT patients (40% vs 9%, p <0.001). Subsequently, patients with VAP had higher $\Delta_{SOFA}$ than those with VAT (mean (SD)

**Table 1. Study population characteristics.**

| | VAT (n = 70) | VAP (n = 136) | p |
|---|---|---|---|
| **Sex** | | | 0.74 |
| Male | 51 (73%) | 102 (75%) | |
| Female | 19 (27%) | 34 (25%) | |
| **Age (years)** | 55.3 (16) | 55 (16.4) | 0.89 |
| **Severity score at ICU admission** | | | |
| SAPS II | 54.6 (19.5) | 60 (16.7) | 0.079 |
| SOFA | 7.2 (4.3) | 8.9 (4) | 0.008 |
| **Admission type** | | | 0.39 |
| Medical | 57 (81%) | 117 (86%) | |
| Surgical | 13 (19%) | 19 (14%) | |
| **Preexisting conditions** | | | |
| COPD | 6 (10%) | 21 (15%) | 0.17 |
| Diabetes mellitus | 14 (20%) | 26 (20%) | 0.88 |
| Immunocompromised patients | 7 (10%) | 23 (17%) | 0.18 |
| Chronic heart disease | 16 (23%) | 16 (12%) | 0.037 |
| Chronic respiratory failure | 3 (4%) | 7 (5%) | 0.78 |
| Cirrhosis | 3 (4%) | 10 (7%) | 0.38 |
| **Previous antibiotic use** | 60 (86%) | 107 (79%) | 0.30 |
| **During ICU stay** | | | |
| Days on mechanical ventilation | 18 (13–29) | 17 (11–29) | 0.23 |
| Days in the ICU | 24 (16–37) | 22 (14–35) | 0.3 |
| ICU mortality | 18 (26%) | 46 (34%) | 0.23 |

Data are presented as number (%) for categorical variables, mean (SD) for normally distributed continuous variables and median (interquartile range) for skewed continuous variables. Admission was defined as surgical if consecutive to a surgery, and medical in the opposite case. *COPD* chronic obstructive pulmonary disease; *SAPS* simplified acute physiology score; *SOFA* sequential organ failure assessment; *VAP* ventilator-associated pneumonia; *VAT* ventilator-associated tracheobronchitis.

**Table 2. Microbiological findings.**

|  | VAT (n = 70) | VAP (n = 136) | p |
|---|---|---|---|
| *Streptococcus pneumoniae* | 2 (3%) | 5 (4%) | > 0.999 |
| *Stenotrophomonas maltophila* | 4 (6%) | 6 (4%) | 0.74 |
| MRSA | 1 (1%) | 1 (1%) | > 0.999 |
| MSSA | 6 (9%) | 21 (15%) | 0.17 |
| *Serratia marcescens* | 3 (4%) | 5 (4%) | > 0.999 |
| *Pseudomonas aeruginosa* | 20 (29%) | 29 (21%) | 0.25 |
| *Proteus mirabilis* | 3 (4%) | 5 (4%) | > 0.999 |
| *Klebsiella pneumoniae* | 9 (13%) | 30 (22%) | 0.11 |
| *Haemophilus influenzae* | 4 (6%) | 6 (4%) | 0.74 |
| *Escherichia coli* | 5 (7%) | 9 (7%) | > 0.999 |
| Enterobacter spp. | 4 (6%) | 21 (15%) | 0.043 |
| *Citrobacter freundii* | 4 (6%) | 1 (1%) | 0.047 |
| *Acinetobacter baumannii* | 0 (0%) | 2 (1%) | 0.55 |

Data are presented as number (%) for categorical variables. *MSSA* methicillin sensitive Staphylococcus aureus; *MRSA* methicillin resistant Staphylococcus aureus; *VAP* ventilator-associated pneumonia; *VAT* ventilator-associated tracheobronchitis.

1 (0.2) vs -0.4 (0.2), p <0.001 (Fig 1). Sepsis criteria according to Sepsis-3 yielded a sensitivity at 0.4 and a specificity at 0.91 for the diagnosis of VAP (Table 3).

Among patients with VA-LRTI, there was no significant difference between survivors and non-survivors in the frequency of sepsis (38% vs 25%, p = 0.076), yet $\Delta_{SOFA}$ was higher in survivors than in non-survivors (median (IQR) 1 (0; 2.5) vs 0 (-1; 2), p = 0.035) (Fig 2). Sepsis criteria according to Sepsis-3 had a sensitivity at 0.38 and a specificity at 0.75 for the prediction of mortality on ICU discharge (Table 3).

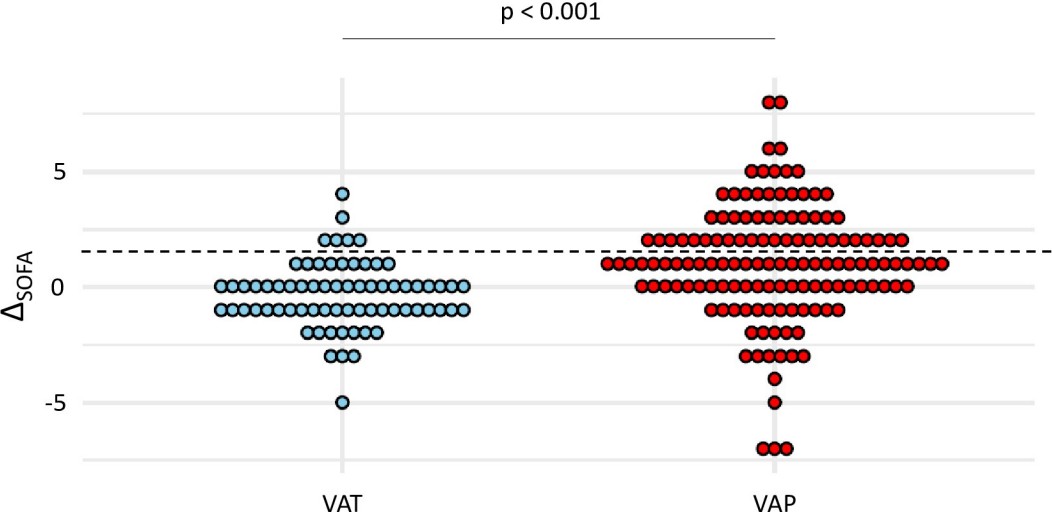

**Fig 1. Dot plots of $\Delta_{SOFA}$ in patients with VAT and VAP.** Values of $\Delta_{SOFA}$ are shown as single dots for each patient. The dash line separates patients with a $\Delta_{SOFA} \geq 2$ from those with a $\Delta_{SOFA} < 2$. *SOFA* sequential organ failure assessment; *VAP* ventilator-associated pneumonia; *VAT* ventilator-associated tracheobronchitis.

**Table 3. Performances of sepsis criteria according to Sepsis-3 for the diagnosis of VAP and the prediction of ICU mortality in patients with VA-LRTI.**

|  | Se | Sp | PPV | NPV | LR+ | LR- |
|---|---|---|---|---|---|---|
| **Diagnosis of VAP** | 0.4 | 0.91 | 0.9 | 0.44 | 4.44 | 0.66 |
| **Prediction of mortality** | 0.38 | 0.75 | 0.4 | 0.73 | 1.48 | 0.84 |

*Se* sensitivity; *Sp* specificity; *PPV* positive predictive value; *NPV* negative predictive value; *LR+* positive likelihood ratio; *LR-* negative likelihood ratio; *VA-LRTI* ventilator-associated lower respiratory tract infection; *VAP* ventilator-associated pneumonia.

## Accuracy of total SOFA, and respiratory SOFA in differentiating VAT from VAP

At the time of VA-LRTI diagnosis, patients with VAP had higher total SOFA than those with VAT (mean (SD) 7.4 (4.4) vs 4.7 (4), p <0.001) (Fig 3). The best performances to differentiate VAT from VAP were observed for a total SOFA ≥ 6. With this cut-off, sensitivity and specificity were found at 0.63 and 0.69, respectively (Table 4).

At the time of VA-LRTI diagnosis, respiratory SOFA was higher in patients with VAP compared to those with VAT (mean (SD) 2.3 (1.2) vs 1.7 (1.1), p = 0.001) (Fig 4). A respiratory SOFA ≥ 3 was associated with the highest Youden index, yielding a sensitivity at 0.49 and a specificity at 0.7 for the diagnosis of VAP (Table 4).

## Accuracy of total SOFA, and respiratory SOFA in predicting mortality

Total SOFA calculated at the time of VA-LRTI diagnosis was found higher in non-survivors than in survivors (mean (SD) 7.8 (5.3) vs 5.9 (3.9), p < 0.01) (Fig 5). The best performances for the prediction of mortality were observed for a total SOFA ≥ 7. With this cut-off, sensitivity and specificity were found at 0.56 and 0.61, respectively (Table 5).

At the time of VA-LRTI diagnosis, respiratory SOFA was higher in non-survivors compared to survivors (mean (SD) 2.4 (1.2) vs 2 (1.1), p = 0.014) (Fig 6). A respiratory SOFA = 4

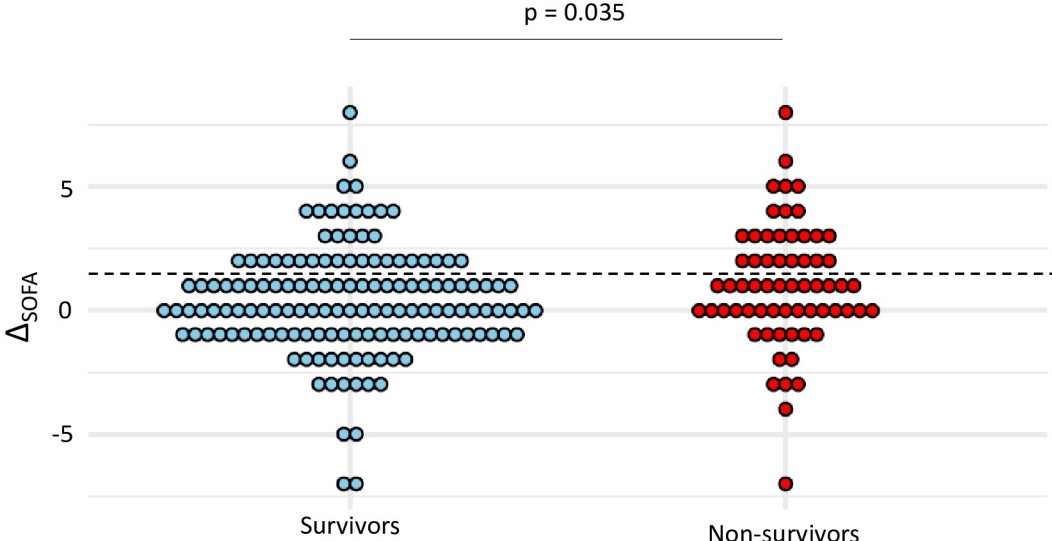

**Fig 2. Dot plots of $\Delta_{SOFA}$ in survivors and non-survivors.** Values of $\Delta_{SOFA}$ are shown as single dots for each patient. The dash line separates patients with a $\Delta_{SOFA} \geq 2$ from those with a $\Delta_{SOFA} < 2$. *SOFA* sequential organ failure assessment; *VAP* ventilator-associated pneumonia; *VAT* ventilator-associated tracheobronchitis.

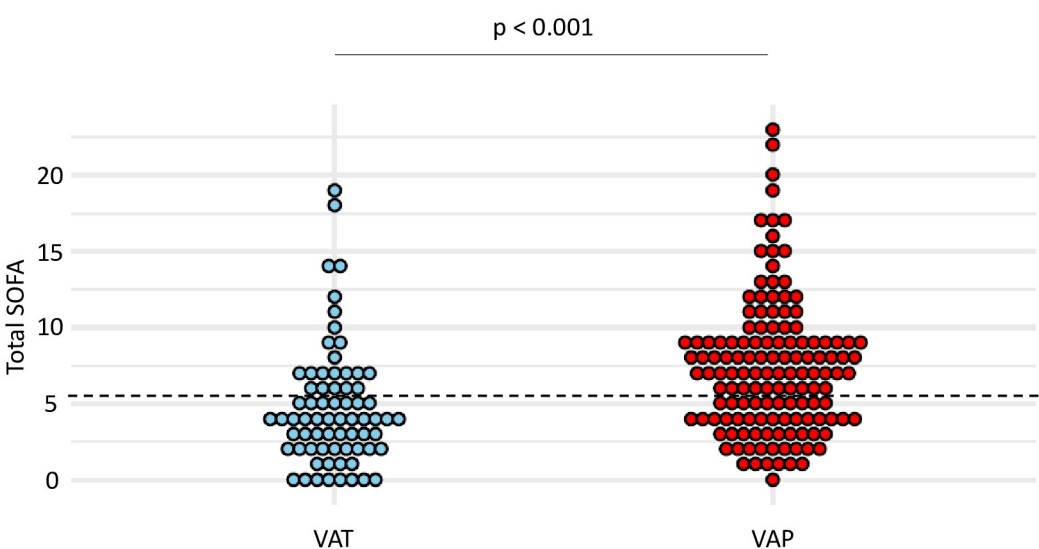

**Fig 3. Dot plots of total SOFA in patients with VAT and VAP.** Values of total SOFA, at VA-LRTI diagnosis, are shown as single dots for each patient. The dash line separates patients with a total SOFA $\geq$ 6 from those with a total SOFA < 6, identified as the best cut-off according to the Youden index. *SOFA* sequential organ failure assessment; *VAP* ventilator-associated pneumonia; *VAT* ventilator-associated tracheobronchitis.

was associated with the highest Youden index, yielding a sensitivity at 0.22 and a specificity at 0.95 for the diagnosis of VAP (Table 5).

## Discussion

The main purpose of this study was to assess whether a diagnosis of sepsis according to the Sepsis-3 definition could be used as a criterion for the early distinction between VAT and VAP and therefore decide whether antibiotic treatment should be started or not. This question may appear of particular significance for the daily clinical practice. First, because early initiation of antibiotic treatment is recommended in VAP, while there is currently no recommendation regarding such a strategy in VAT [2]. Therefore, finding reliable criteria for the early distinction between VAT and VAP seems relevant. Furthermore, physicians may experience difficulties in making the early distinction between VAT and VAP in the daily clinical practice, due to

**Table 4. Performances of total SOFA and respiratory SOFA for the diagnosis of VAP in patients with VA-LRTI.**

|  | Se | Sp | PPV | NPV | LR+ | LR- | Youden index |
|---|---|---|---|---|---|---|---|
| **Total SOFA** | | | | | | | |
| $\geq$ **6** | 0.63 | 0.69 | 0.79 | 0.48 | 1.99 | 0.50 | 0.31 |
| $\geq$ **7** | 0.55 | 0.76 | 0.82 | 0.46 | 2.27 | 0.44 | 0.3 |
| $\geq$ **8** | 0.45 | 0.86 | 0.86 | 0.44 | 3.14 | 0.32 | 0.3 |
| **Respiratory SOFA** | | | | | | | |
| $\geq$ **1** | 0.94 | 0.21 | 0.70 | 0.65 | 1.20 | 0.83 | 0.16 |
| $\geq$ **2** | 0.75 | 0.40 | 0.71 | 0.45 | 1.25 | 0.80 | 0.15 |
| $\geq$ **3** | 0.49 | 0.70 | 0.76 | 0.41 | 1.62 | 0.62 | 0.19 |
| = **4** | 0.13 | 0.94 | 0.81 | 0.36 | 2.19 | 0.46 | 0.07 |

*SOFA* sequential organ failure assessment; *Se* sensitivity; *Sp* specificity; *PPV* positive predictive value; *NPV* negative predictive value; *LR+* positive likelihood ratio; *LR-* negative likelihood ratio; *VA-LRTI* ventilator-associated lower respiratory tract infection; *VAP* ventilator-associated pneumonia.

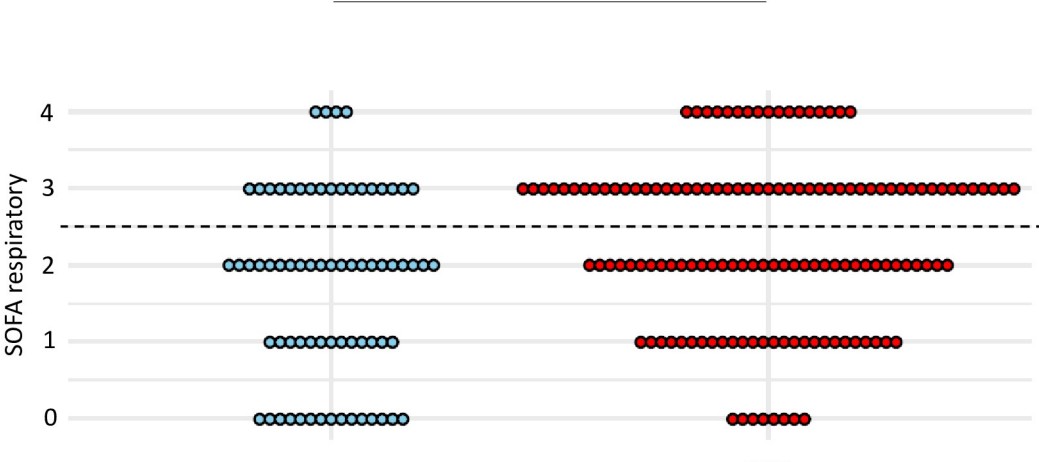

**Fig 4. Dot plots of respiratory SOFA in patients with VAT and VAP.** Values of respiratory SOFA, at VA-LRTI diagnosis, are shown as single dots for each patient. The dash line separates patients with a respiratory SOFA $\geq$ 3 from those with a respiratory SOFA < 3, identified as the best cut-off according to the Youden index. *SOFA* sequential organ failure assessment; *VAP* ventilator-associated pneumonia; *VAT* ventilator-associated tracheobronchitis.

confounding factors on chest X-ray images (pleural effusion, lung edema) [4, 5] or because of the frequent delay in appearance of lung opacities [6]. Additionally, data from the TAVeM study show that patients with VAP experience greater severity of illness than those with VAT, thus suggesting that a diagnosis of sepsis, reflecting a worsening in organ failures, could be found more frequently in VAP compared to VAT.

Our results suggest that criteria for sepsis according to the Sepsis-3 definition had a high specificity and a low sensitivity to distinguish VAT from VAP, but only a moderate specificity

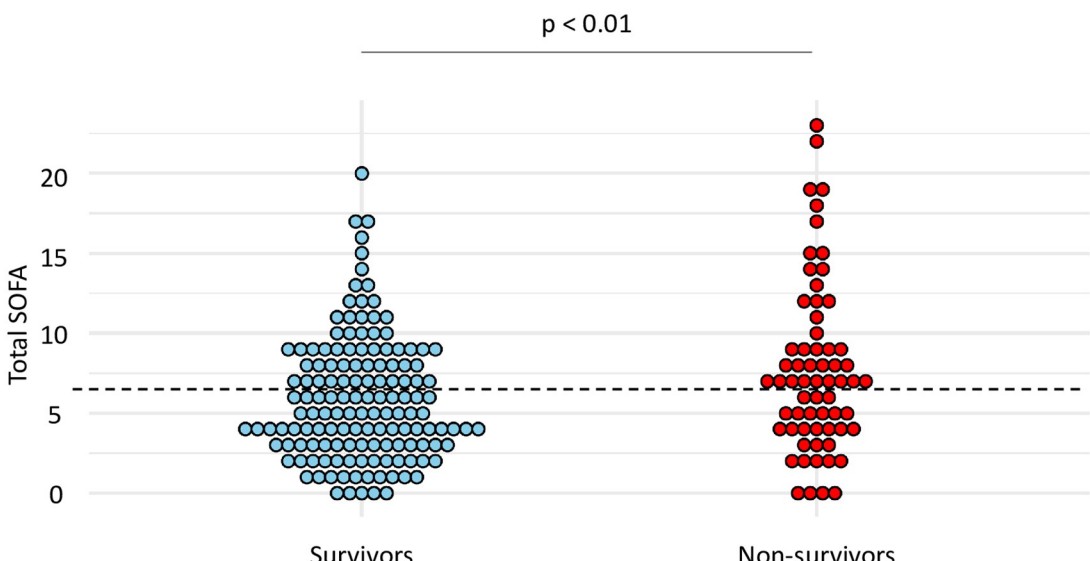

**Fig 5. Dot plots of total SOFA in survivors and non-survivors.** Values of total SOFA are shown as single dots for each patient. The dash line separates patients with a total SOFA $\geq$ 7 from those with a total SOFA < 7. *SOFA* sequential organ failure assessment.

**Table 5. Performances of total SOFA and respiratory SOFA for the prediction of mortality in patients with VA-LRTI.**

|  | Se | Sp | PPV | NPV | LR+ | LR- | Youden index |
|---|---|---|---|---|---|---|---|
| **Total SOFA** | | | | | | | |
| ≥ 5 | 0.7 | 0.44 | 0.36 | 0.76 | 1.26 | 0.79 | 0.15 |
| ≥ 6 | 0.61 | 0.52 | 0.36 | 0.75 | 1.27 | 0.79 | 0.13 |
| ≥ 7 | 0.56 | 0.61 | 0.39 | 0.75 | 1.43 | 0.7 | 0.17 |
| **Respiratory SOFA** | | | | | | | |
| ≥ 1 | 0.91 | 0.12 | 0.32 | 0.74 | 1.03 | 0.97 | 0.03 |
| ≥ 2 | 0.77 | 0.33 | 0.34 | 0.76 | 1.14 | 0.87 | 0.10 |
| ≥ 3 | 0.52 | 0.62 | 0.38 | 0.74 | 1.36 | 0.74 | 0.14 |
| = 4 | 0.22 | 0.95 | 0.67 | 0.73 | 4.44 | 0.23 | 0.17 |

*SOFA* sequential organ failure assessment; *Se* sensitivity; *Sp* specificity; *PPV* positive predictive value; *NPV* negative predictive value; *LR+* positive likelihood ratio; *LR-* negative likelihood ratio; *VA-LRTI* ventilator-associated lower respiratory tract infection.

and a low sensitivity to predict ICU-mortality in patients with confirmed VA-LRTI. Further, total SOFA ≥ 6, identified as the best cut-off, showed moderate sensitivity and specificity to differentiate VAT from VAP. On the other hand, the best performances of respiratory SOFA to make the distinction between VAT and VAP were reached for a cut-off ≥ 3. With this threshold, respiratory SOFA had a low sensitivity and a moderate specificity for the diagnosis of VAP.

Our study highlights the weak accuracy of the Sepsis-3 criteria for diagnosis of sepsis in distinguishing VAT from VAP and in predicting mortality in patients with VA-LRTI. Importantly, our results underline the poor sensitivity and negative predictive value of these severity criteria to rule out a diagnosis of VAP, and therefore decide that antibiotic treatment should not be started. These latter findings may be analyzed in light of the data previously published in the literature, which suggest a greater severity of illness in VAP than in VAT. This was reported in a worldwide prospective cohort of ventilated patients by the TAVeM study group,

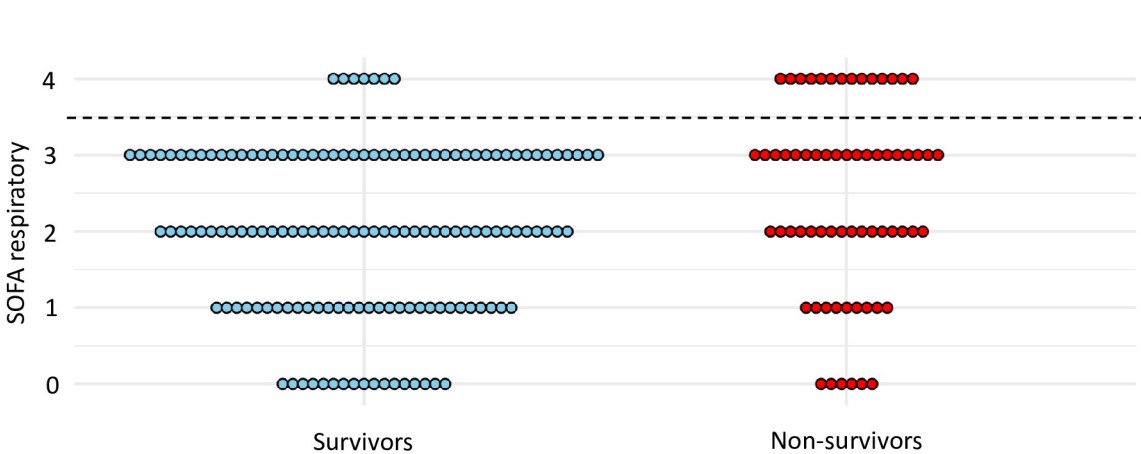

**Fig 6. Dot plots of respiratory SOFA in survivors and non-survivors.** Values of respiratory SOFA are shown as single dots for each patient. The dash line separates patients with a respiratory SOFA = 4 from those with a respiratory SOFA < 4. *SOFA* sequential organ failure assessment.

which found a higher mean SOFA value in VAP than in VAT, and subsequently identified VAP as an independent risk factor of mortality, conversely to VAT [1]. Moreover, in this study, the authors reported similar SOFA scores on baseline for patients with VAT and VAP, thus conveying the idea that organ failures were worsening in VAP but not in VAT. However, our study highlights the notable overlap in $\Delta_{\text{SOFA}}$ when comparing patients with VAP to those with VAT, as well as between survivors and non-survivors, explaining its poor accuracy as a diagnostic and prognostic tool in the daily clinical practice.

Additionally, we found low performances of total SOFA and respiratory SOFA in distinguishing VAT from VAP in our study. However, our results show a higher respiratory SOFA in patients with VAP compared to those with VAT, consistently with previous findings from the TAVeM cohort. Indeed, in that study, patients with VAP experienced more frequent episodes of hypoxemia than in VAT. This finding suggests that a higher respiratory SOFA might be used to distinguish VAP from VAT. However, this hypothesis was not supported by our findings, because of the notable overlap in respiratory SOFA values between the two groups. This result may also be analyzed in regards of the criteria issued by the CDC for the diagnosis of ventilator-associated events, based on the presence of gas exchange worsening [10]. Even though developed for a purpose of reproducibility, these criteria seem to show a poor agreement with the classical definition of VAP, as outlined by several studies [7, 11, 12]. Our results, showing that severity of hypoxemia exhibited poor accuracy for the diagnosis of VAP tend to support these data.

Other diagnosis tools have been proposed to make the early distinction between VAT and VAP. The usefulness of CRP and PCT was thus investigated by the TAVeM study group, who reported a marked overlap in CRP and PCT concentrations between patients with VAT and those with VAP [13]. Subsequently, the area under the ROC curve was found at 0.6 (95% confidence interval (CI) 0.54–0.65) for CRP and 0.63 (95% CI 0.57–0.7) for PCT, reflecting the poor accuracy of these criteria in differentiating VAT from VAP. Furthermore, several studies have underlined the potential interest of alternatives to chest X-ray to allow a more accurate diagnosis of VA-LRTI. This was notably highlighted by Self *et al.* who reported a better sensitivity of computed tomography (CT) compared to chest X-ray for the detection of lung opacities, raising the question of using this technique to improve the detection of VAP [14]. However, the use of CT to differentiate VAT from VAP can hardly be proposed for daily clinical practice, because of greater costs, time consumption, and higher risks associated with intra-hospital transports in critically ill patients [15, 16]. On the other hand, lung ultrasound has been proposed as an alternative for the diagnosis of VAP, and might be a promising technique in a near future [17, 18]. However, this operator-dependent technique might suffer from a lack of reproducibility and still needs to be validated in larger cohorts of patients. In another approach, Martin-Loeches and Pobo suggested that bronchoscopy might be a helpful tool by identifying secretions coming from deep lung regions, which would be found in VAP but not in VAT [19]. However, no study to date has evaluated this strategy and the usefulness of bronchoscopy in distinguishing VAT from VAP remains unclear. Altogether, these data stress the need for further studies to improve the early detection of VAP in patients with microbiologically confirmed VA-LRTI.

Our study has several limitations. First, this study was conducted in a single center, therefore limiting the applicability of our results to the general population of critically ill patients. However, population characteristics were in line with those observed in the TAVeM study, which enrolled patients from 114 ICU worldwide [1]. Further, we performed a retrospective analysis. Nevertheless, consequences on our results were likely very limited, as we had no missing data for the variables included in our analysis. Additionally, continuous surveillance of ICU-acquired infections allowed prospective identification of patients with VA-LRTI. Another

limitation of our study lays in the exclusive use of chest X-ray rather than CT-scan to distinguish VAT from VAP. This was probably associated with a lack of sensitivity in detecting lung opacities in some patients with VAP [4, 5], thus raising concerns about the relevance of using this exam for the diagnosis of VAP in our study. However, it should be remembered that diagnosis of VAT and VAP was made with full knowledge of the patient's medical record. This included the possibility of observing the appearance of prior opacities on chest X-ray, or secondarily unmasking a differential diagnosis of VAP, such as cardiac overload, atelectasis, or pleural effusion. In these different cases, the diagnosis of VAP could reasonably be made ultimately on the basis of the chest X-ray, while still being difficult in daily practice at the patient's bedside. Therefore, chest X-ray, which was used in the largest international study describing the characteristics of VAT and VAP, seemed relevant in our study [1]. Moreover, our definitions of VAT and VAP are consistent with those in which a difference in mortality rates were reported [1]. Furthermore, it may be reminded that use of CT-scan for the diagnosis of VAP remains limited in the daily clinical practice, due to greater cost, increased time required to obtain images, higher radiation exposure and risks associated with intra-hospital transportation [20]. As a result, CT scan was only used in a minority of cases to diagnose VAP in our patients, and therefore could not be used in our study. This situation reflects the low practical use of this tool to diagnose VAP. Thus, although the potential value of CT scan in a comprehensive and mechanistic study may be acknowledged, its practical use for the diagnosis of VAP is likely to be limited. Higher levels of SOFA and SAPS2 at ICU admission in VAP compared to VAT was another limitation of our study, as these may partly explain further worsening of organ failures in patients with VAP. Therefore, one could argue that this difference between groups was not addressed in our analysis. However, the goal of our study was merely diagnostic. Accordingly, the practical question that our study aimed to answer is whether a worsening in organ failures could be used as a reliable criterion to differentiate VAT from VAP, and could therefore be used in the daily clinical practice to decide whether antibiotic treatments should be started or not. To answer this practical question, it appears that ΔSOFA must be assessed as it would by the physician at the patient's bedside, without adjustment for potential confounders. Last, our study did not compare timings of diagnosis of VAP when comparing severity criteria including sepsis criteria according to the Sepsis-3 definition vs chest X-ray. However, our study was not designed to answer this question, but merely aimed to evaluate severity criteria at the time of clinical evidence of VA-LRTI for the diagnosis of VAP, and may thus bring relevant information regarding this practical question at the patient's bedside.

## Conclusions

In patients with evidence of VA-LRTI, sepsis criteria following the Sepsis-3 definition exhibit low performances for the diagnosis of VAP and for the prediction of mortality. Total SOFA and respiratory SOFA were also inaccurate in differentiating VAT from VAP. Accordingly, our results do not support the use of these criteria to drive the early initiation of antibiotic treatments in patients with VA-LRTI.

## Supporting information

**S1 Fig. ROC curves of $\Delta_{SOFA}$ (A), total SOFA (B), and respiratory SOFA (C) for the diagnosis of VAP in patients with VA-LRTI.** *SOFA* sequential organ failure assessment.
(TIF)

**S2 Fig. ROC curves of $\Delta_{SOFA}$ (A), total SOFA (B), and respiratory SOFA (C) for the prediction of mortality on ICU discharge in patients with VA-LRTI.** *SOFA* sequential organ failure assessment.
(TIF)

**S1 Appendix. Criteria for diagnosis of VAT and VAP.**
(DOCX)

**S2 Appendix. Sepsis-3 criteria for diagnosis of sepsis and septic shock.**
(DOCX)

## Author Contributions

**Conceptualization:** Alexandre Gaudet, Saad Nseir.

**Formal analysis:** Alexandre Gaudet, Matthieu Devos, Saad Nseir.

**Investigation:** Alexandre Gaudet, Matthieu Devos, Sylvain Keignart, Olivier Pouly, Sylvain Lecailtel, Frédéric Wallet, Saad Nseir.

**Methodology:** Saad Nseir.

**Supervision:** Saad Nseir.

**Validation:** Saad Nseir.

**Writing – original draft:** Alexandre Gaudet.

**Writing – review & editing:** Saad Nseir.

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
