## [Decision Letter · Decision Letter 0]

26 Oct 2020

PONE-D-20-26296

Usefulness of Sepsis-3 in diagnosing and predicting mortality of ventilator-associated lower respiratory tract infections

PLOS ONE

Dear Dr. Gaudet,

Thank you for submitting your manuscript to PLOS ONE. After careful consideration, we feel that it has merit but does not fully meet PLOS ONE’s publication criteria as it currently stands. Therefore, we invite you to submit a revised version of the manuscript that addresses the points raised during the review process.

Overall, well written and some aspects are paid our attention. However, several points need to be modified. Please see our reviewers comments below. 

We look forward to receiving your revised manuscript.

Kind regards,

Yutaka Kondo

Academic Editor

PLOS ONE

Journal Requirements:

'I have read the journal's policy and the authors of this manuscript have the following competing interests:SN received fees from MSD, Pfizer, Gilead, BioMérieux, and Bio Rad for lectures.

Other authors declare that they have no competing interest.'

a. Please confirm that this does not alter your adherence to all PLOS ONE policies on sharing data and materials, by including the following statement: "This does not alter our adherence to  PLOS ONE policies on sharing data and materials.” (as detailed online in our guide for authors http://journals.plos.org/plosone/s/competing-interests).  If there are restrictions on sharing of data and/or materials, please state these.

Please note that we cannot proceed with consideration of your article until this information has been declared.

Reviewers' comments:

Reviewer's Responses to Questions

**Comments to the Author**

1. Is the manuscript technically sound, and do the data support the conclusions?

Reviewer #1: Yes

Reviewer #2: No

Reviewer #3: No

2. Has the statistical analysis been performed appropriately and rigorously? 

Reviewer #1: Yes

Reviewer #2: Yes

Reviewer #3: Yes

3. Have the authors made all data underlying the findings in their manuscript fully available?

Reviewer #1: Yes

Reviewer #2: Yes

Reviewer #3: Yes

4. Is the manuscript presented in an intelligible fashion and written in standard English?

Reviewer #1: Yes

Reviewer #2: Yes

Reviewer #3: Yes

5. Review Comments to the Author

Reviewer #1: This retrospective study evaluated diagnostic ability of sepsis-3 criteria, total SOFA and respiratory SOFA to differentiate VAP from VAT, and concluded that sepsis-3 was not useful. It is worth publishing, since logic looks appropriate and conclusion is reasonable. However, it also includes several problems to be addressed.

P6, L90

“All adults patients with a diagnosis of VA-LRTI were retrospectively included in the study. The diagnosis of VA-LRTI was prospectively performed by ICU physicians.”

If authors regarded diagnosis was made prospectively, they had to review not only patients diagnosed as VA-LRTI, but also all patients admitted to the ICU during study period. If they had reviewed only retrospectively-included VALRTI patients, prospective diagnosis of VALRTI could not be possible. Or did they only diagnose VAT or VAP among VALRTI patients? Please explain and describe clearly in the Methods section.

P7, L120

Regarding Objective of this study, “accuracy of predicting ICU mortality” was evaluated only in sepsis-3 analysis, not in analyses of total SOFA and respiratory SOFA. However, since authors repeatedly compared sepsis-3 to SOFA in the context of this study, they should have evaluated SOFA’s ability in predicting mortality. Otherwise, they should focus only on ability of diagnosis of VAT and VAP in comparison between sepsis-3 and SOFA, and eliminate results of mortality in spsis-3 analysis.

Reviewer #2: The definition was vague. Did the authors look for signs of pulmonary consolidation in admission? no CT scan?

What is category of admission? Please explain details.

The adjusting the severity is important.

From reading the paper many things remain unclear. What is special about this study in relation to future strategy or research? The study lacks clarity and furthermore, it does not become clear what is particulary special and what the readers can learn from it.

Reviewer #3: I greatly appreciate giving me the opportunity to review the manuscript entitled “Usefulness of Sepsis-3 in diagnosing and predicting mortality of ventilator-associated lower respiratory tract infections”. It is true that the more accurate and rapid diagnostic tool to distinguish VAP and VAT is necessary to improve the management of critically ill ventilated patients. However, the aim of the current study is not clear, and the study design seems not optimal. My concerns are as follows.

1. First, the use of the term “sepsis-3 criteria” is not appropriate. Sepsis-3 is just “Sepsis-3, i.e., it is used to define sepsis based on progress of organ dysfunctions, and it is not a general diagnostic tool for other diseases. Therefore, the authors must use the term such as “increase in SOFA score”, not “sepsis-3 criteria”.

2. The aim of the study is unclear. If the authors have hypothesized that the use of SOFA score can diagnose the progress of infection to lung parenchyma (the true definition of VAP) more accurately compared with chest X-ray, the gold-standard diagnosis must be established by more accurate modality such as chest CT, and the accuracy of two diagnostic methods; SOFA vs. chest X-ray, must be evaluated. Or if the authors are willing to demonstrate the use of SOFA score can diagnose gradual VAP more rapidly compared with chest X-ray, they must show the timing of diagnosis of VAP by two diagnostic methods. In the current manuscript, the authors have just compared the SOFA scores between VAP and VAT patients diagnosed by chest X-ray. It is not clear how they want to utilize the SOFA score to distinguish VAP and VAT.

6. PLOS authors have the option to publish the peer review history of their article (what does this mean?). If published, this will include your full peer review and any attached files.

Reviewer #1: No

Reviewer #2: No

Reviewer #3: **Yes: **Kentaro Tojo

---

## [Author Response · Author response to Decision Letter 0]

24 Nov 2020

Dear Editors, 

Dear Reviewers, 

Thank you for your helpful suggestions regarding our manuscript. 

Following your comments, we have provided some additional information which, we believe, should improve the quality of the article. 

Please find appended below our point-by-point response to your questions: 

Reviewer #1: This retrospective study evaluated diagnostic ability of sepsis-3 criteria, total SOFA and respiratory SOFA to differentiate VAP from VAT, and concluded that sepsis-3 was not useful. It is worth publishing, since logic looks appropriate and conclusion is reasonable. However, it also includes several problems to be addressed.

1. P6, L90 “All adults patients with a diagnosis of VA-LRTI were retrospectively included in the study. The diagnosis of VA-LRTI was prospectively performed by ICU physicians.” If authors regarded diagnosis was made prospectively, they had to review not only patients diagnosed as VA-LRTI, but also all patients admitted to the ICU during study period. If they had reviewed only retrospectively-included VALRTI patients, prospective diagnosis of VALRTI could not be possible. Or did they only diagnose VAT or VAP among VALRTI patients? Please explain and describe clearly in the Methods section.

Thank you for pointing out this lack of clarity in our initial formulation. Physicians in charge of the patients identified VA-LRTI episodes as they occurred (prospective diagnosis), and retrospectively collected data on these patients. From a methodological point of view, this is therefore a retrospective study. Accordingly, and to avoid confusion, we have changed the description of the study design in the methods section into: “Continuous surveillance of ICU-acquired infections allowed prospective identification of patients with VA-LRTI. These patients were subsequently included in this retrospective study and other data were extracted from electronic files”.

2. P7, L120 Regarding Objective of this study, “accuracy of predicting ICU mortality” was evaluated only in sepsis-3 analysis, not in analyses of total SOFA and respiratory SOFA. However, since authors repeatedly compared sepsis-3 to SOFA in the context of this study, they should have evaluated SOFA’s ability in predicting mortality. Otherwise, they should focus only on ability of diagnosis of VAT and VAP in comparison between sepsis-3 and SOFA, and eliminate results of mortality in spsis-3 analysis.

We fully agree with your comment, and have therefore completed the manuscript by reporting the assessment of total SOFA and respiratory SOFA in predicting mortality in main text, Table 5, Figures 5 and 6, and Figure S2.

Reviewer #2: 

1. The definition was vague. Did the authors look for signs of pulmonary consolidation in admission? no CT scan?

Thank you for pointing out these important matters. To avoid any confusion for the reader, we now explicitly mention in the manuscript that criteria from the International ERS/ESICM/ESCMID/ALAT guidelines for the management of hospital-acquired pneumonia and ventilator-associated pneumonia were used for the definition of VAT and VAP (1). Furthermore, these criteria are now described in S1 Appendix, in addition to criteria detailed in the methods section.

This definition does not specify whether chest X-ray or CT-scan should be used for the diagnosis of VA-LRTI. In our study, only chest X-ray was used to distinguish VAT from VAP as CT-scan is not routinely performed for the diagnosis of VAP, due to greater cost, increased time required to obtain images, higher radiation exposure and risks associated with intra-hospital transportation (2).

We have therefore completed the methods section of our manuscript, by explicitly stating that CT-scan was not used to differentiate VAT from VAP in our study.

We acknowledge that lung opacities were probably missed in some patients with VAP, given the reported lack of sensitivity of chest X-ray for the detection of pneumonia (3,4). However, the use of chest X-ray allowed us to diagnose VAT and VAP using the same criteria than in the TAVeM study. Accordingly, our definitions of VAP and VAT are consistent with those in which a difference in mortality rates were reported (5). 

These points are now mentioned in the limitations section of the manuscript. 

2. What is category of admission? Please explain details.

We acknowledge that clarifications were needed regarding definitions of admission categories.

We are actually referring to the type of admission in ICU. Admission were defined as surgical if consecutive to a surgery, and medical in the opposite case. 

The term “Category of admission” has been replaced by “Admission type”, and these definitions are now specified in caption of Table 1.

3. The adjusting the severity is important.

We thank you for raising the important question of adjusting on severity in our analysis.

As pointed out in your comment, total SOFA at ICU admission was significantly higher in VAP than in VAT. Assessment of SAPS2 at ICU admission also showed a trend for higher severity in VAP. 

We perfectly acknowledge that confounding factors such as higher levels of severity at ICU admission may partly explain further increases in ∆SOFA in patients with VAP. Addressing this issue would be crucial from the prospect of a study aiming to understand the mechanisms associated with the development of VAP.

However, we’d like to remind that the goal of our study was merely diagnostic. Accordingly, the practical question that our study aimed to answer is whether a worsening in organ failures could be used as a reliable criterion to differentiate VAT from VAP, and could therefore be used in the daily clinical practice to decide whether to initiate or not antibiotic treatments. To answer this practical question, it appears that severity criteria, including ∆SOFA, must be assessed as they would by the physician at the patient’s bedside, without adjustment for potential confounders.

Initial version of our manuscript lacked clarity regarding these points, which have now been developed in the discussion section of the revised version. We believe it will help the reader to understand more clearly the purpose of our study.

4. From reading the paper many things remain unclear. What is special about this study in relation to future strategy or research? The study lacks clarity and furthermore, it does not become clear what is particulary special and what the readers can learn from it.

The rationale of our study mainly relies on the 3 following points: 

a) Early initiation of antibiotic treatment is recommended in VAP, while there is currently no recommendation regarding such a strategy in VAT (6). Therefore, finding reliable criteria for the early distinction between VAT and VAP seems relevant.

b) Physicians may experience difficulties in making the early distinction between VAT and VAP in the daily clinical practice, due to confounding factors on chest X-ray images (pleural effusion, lung edema) (3,4) or because of the frequent delay in appearance of lung opacities (7).

c) Data from the TAVeM study show that patients with VAP experience greater severity of illness than those with VAT, thus suggesting that a diagnosis of sepsis, reflecting a worsening in organ failures, could be found more frequently in VAP compared to VAT.

Therefore, our study aimed to clarify a practical question for the daily clinical practice, i.e. whether diagnosis of sepsis according to Sepsis-3 criteria, reflecting a greater severity of illness, would be a reliable criterion to diagnose VAP in patients with VA-LRTI, and accordingly, drive the early initiation of antibiotic treatment. 

Our results suggest that assessment of severity, especially through Sepsis-3 criteria, exhibit poor performances to early differentiate VAT from VAP. Importantly, we found poor sensitivity and negative predictive value of these criteria to rule out a diagnosis of VAP. As a consequence, these criteria should probably not be used to drive the early initiation of antimicrobial therapy in patients with evidence of VA-LRTI. Especially, the absence of diagnosis of sepsis appears as a bad criterion to decide not to initiate antibiotic treatments. 

To the best of our knowledge, our study is the first to address this question, and we believe it could have a practical impact for the physician at the patient’s bedside.

However, we acknowledge that our message was not enough clearly developed in our manuscript. Accordingly, these points are now clearly developed in the first part of our discussion. We hope these changes will help bringing more clearly the aims and results of our study to the reader.

Reviewer #3: I greatly appreciate giving me the opportunity to review the manuscript entitled “Usefulness of Sepsis-3 in diagnosing and predicting mortality of ventilator-associated lower respiratory tract infections”. It is true that the more accurate and rapid diagnostic tool to distinguish VAP and VAT is necessary to improve the management of critically ill ventilated patients. However, the aim of the current study is not clear, and the study design seems not optimal. My concerns are as follows.

1. First, the use of the term “sepsis-3 criteria” is not appropriate. Sepsis-3 is just “Sepsis-3, i.e., it is used to define sepsis based on progress of organ dysfunctions, and it is not a general diagnostic tool for other diseases. Therefore, the authors must use the term such as “increase in SOFA score”, not “sepsis-3 criteria”.

We acknowledge that the term “Sepsis-3 criteria” was not used in an appropriate way in our manuscript. Especially, use of this term may be confusing as it does not indicate whether criteria for sepsis or septic shock were evaluated. However, the main purpose of our study was to assess whether the presence of sepsis would be a good marker of VAP, and conversely, if absence of sepsis would be sufficient to rule out VAP, and therefore decide not to initiate antibiotic treatment. As a consequence, we believe it is important for a matter of clarity to keep reminding to the reader that our study evaluated the diagnostic criteria for sepsis. 

Therefore, we have replaced the term “Sepsis-3 criteria” by “sepsis criteria according to the Sepsis-3 definition” throughout the article. Furthermore, description of criteria for sepsis and septic shock according to the Sepsis-3 definition has been added in S2 Appendix.

2. The aim of the study is unclear. If the authors have hypothesized that the use of SOFA score can diagnose the progress of infection to lung parenchyma (the true definition of VAP) more accurately compared with chest X-ray, the gold-standard diagnosis must be established by more accurate modality such as chest CT, and the accuracy of two diagnostic methods; SOFA vs. chest X-ray, must be evaluated. Or if the authors are willing to demonstrate the use of SOFA score can diagnose gradual VAP more rapidly compared with chest X-ray, they must show the timing of diagnosis of VAP by two diagnostic methods. In the current manuscript, the authors have just compared the SOFA scores between VAP and VAT patients diagnosed by chest X-ray. It is not clear how they want to utilize the SOFA score to distinguish VAP and VAT.

We fully agree on the importance of clearly describing the purpose of our study. 

Our study aimed to clarify a practical question for the daily clinical practice, i.e. whether diagnosis of sepsis according to Sepsis-3 criteria, reflecting a greater severity of illness, would be a reliable criterion to early detect or rule out VAP, and therefore decide whether to initiate or not antibiotic treatment in VA-LRTI. In the prospect of a practical diagnostic study, this assessment was made by the time of appearance of clinical signs of VA-LRTI, when radiographical signs of pneumonia on chest X-ray are frequently hidden or delayed (2–4,7).

In our study, only chest X-ray was used to distinguish VAT from VAP as CT-scan is not routinely performed for the diagnosis of VAP, due to greater cost, increased time required to obtain images, higher radiation exposure and risks associated with intra-hospital transportation2.

We acknowledge that lung opacities were probably missed in some patients with VAP, given the reported lack of sensitivity of chest X-ray for the detection of pneumonia (3,4). However, the use of chest X-ray allowed us to diagnose VAT and VAP using the same criteria than in the TAVeM study. Accordingly, our definitions of VAP and VAT are consistent with those in which a difference in mortality rates were reported (5). 

Our results suggest that assessment of severity, especially through sepsis criteria according to the Sepsis-3 definition, exhibit poor performances to early differentiate VAT from VAP. Importantly, we found poor sensitivity and negative predictive value of these criteria to rule out a diagnosis of VAP. As a consequence, these criteria should probably not be used to drive the early initiation of antimicrobial therapy in patients with evidence of VA-LRTI. Especially, the absence of diagnosis of sepsis appears as a bad criterion to decide not to initiate antibiotic treatments.

Finally, we agree on the interest of comparing timings of diagnosis of VAP when using both methods, i.e. comparing severity criteria vs chest X-ray. However, our study was not designed to answer this question, but merely aimed to evaluate severity criteria at the time of clinical evidence of VA-LRTI for the diagnosis of VAP, which corresponds to a practical question at the patient’s bedside.

All these points have been developed in the discussion and limitations of the revised manuscript. We believe it will help the reader to understand more clearly the purpose of our study, as well as the significance of our results for the daily clinical practice. 

References:

1. Torres A, Niederman MS, Chastre J, Ewig S, Fernandez-Vandellos P, Hanberger H, et al. International ERS/ESICM/ESCMID/ALAT guidelines for the management of hospital-acquired pneumonia and ventilator-associated pneumonia: Guidelines for the management of hospital-acquired pneumonia (HAP)/ventilator-associated pneumonia (VAP) of the European Respiratory Society (ERS), European Society of Intensive Care Medicine (ESICM), European Society of Clinical Microbiology and Infectious Diseases (ESCMID) and Asociación Latinoamericana del Tórax (ALAT). Eur Respir J. 2017 Sep;50(3). 

2. Keane S, Vallecoccia MS, Nseir S, Martin-Loeches I. How Can We Distinguish Ventilator-Associated Tracheobronchitis from Pneumonia? Clin Chest Med. 2018;39(4):785–96. 

3. Butler KL, Sinclair KE, Henderson VJ, McKinney G, Mesidor DA, Katon-Benitez I, et al. The chest radiograph in critically ill surgical patients is inaccurate in predicting ventilator-associated pneumonia. Am Surg. 1999 Sep;65(9):805–9; discussion 809-810. 

4. Graat ME, Choi G, Wolthuis EK, Korevaar JC, Spronk PE, Stoker J, et al. The clinical value of daily routine chest radiographs in a mixed medical-surgical intensive care unit is low. Crit Care Lond Engl. 2006 Feb;10(1):R11. 

5. Martin-Loeches I, Povoa P, Rodríguez A, Curcio D, Suarez D, Mira J-P, et al. Incidence and prognosis of ventilator-associated tracheobronchitis (TAVeM): a multicentre, prospective, observational study. Lancet Respir Med. 2015;3(11):859–68. 

6. Kalil AC, Metersky ML, Klompas M, Muscedere J, Sweeney DA, Palmer LB, et al. Management of Adults With Hospital-acquired and Ventilator-associated Pneumonia: 2016 Clinical Practice Guidelines by the Infectious Diseases Society of America and the American Thoracic Society. Clin Infect Dis Off Publ Infect Dis Soc Am. 2016 Sep 1;63(5):e61–111. 

7. Ramirez P, Lopez-Ferraz C, Gordon M, Gimeno A, Villarreal E, Ruiz J, et al. From starting mechanical ventilation to ventilator-associated pneumonia, choosing the right moment to start antibiotic treatment. Crit Care Lond Engl. 2016 Jun 3;20(1):169.

---

## [Decision Letter · Decision Letter 1]

17 Dec 2020

PONE-D-20-26296R1

Usefulness of Sepsis-3 in diagnosing and predicting mortality of ventilator-associated lower respiratory tract infections

PLOS ONE

Dear Dr. Gaudet,

Thank you for submitting your manuscript to PLOS ONE. After careful consideration, we feel that it has merit but does not fully meet PLOS ONE’s publication criteria as it currently stands. Therefore, we invite you to submit a revised version of the manuscript that addresses the points raised during the review process.

ACADEMIC EDITOR: Our reviewers have evaluated your revised manuscript. Please see our reviewers comments and revise as much as possible. At least, the authors need to state the points. Thank you very much. 

We look forward to receiving your revised manuscript.

Kind regards,

Yutaka Kondo

Academic Editor

PLOS ONE

Reviewers' comments:

Reviewer's Responses to Questions

**Comments to the Author**

1. If the authors have adequately addressed your comments raised in a previous round of review and you feel that this manuscript is now acceptable for publication, you may indicate that here to bypass the “Comments to the Author” section, enter your conflict of interest statement in the “Confidential to Editor” section, and submit your "Accept" recommendation.

Reviewer #1: All comments have been addressed

Reviewer #2: All comments have been addressed

Reviewer #3: (No Response)

2. Is the manuscript technically sound, and do the data support the conclusions?

Reviewer #1: (No Response)

Reviewer #2: Yes

Reviewer #3: No

3. Has the statistical analysis been performed appropriately and rigorously? 

Reviewer #1: (No Response)

Reviewer #2: Yes

Reviewer #3: Yes

4. Have the authors made all data underlying the findings in their manuscript fully available?

Reviewer #1: (No Response)

Reviewer #2: Yes

Reviewer #3: Yes

5. Is the manuscript presented in an intelligible fashion and written in standard English?

Reviewer #1: (No Response)

Reviewer #2: Yes

Reviewer #3: Yes

6. Review Comments to the Author

Reviewer #1: (No Response)

Reviewer #2: Excellent work addressing previous concerns. Happy with the responses to my questions and the revisions.

Reviewer #3: Thank you very much for giving me the opportunity to review the revised version of the manuscript entitled “Usefulness of Sepsis-3 in diagnosing and predicting mortality of ventilator-associated lower respiratory tract infections”. I have understood the aim of the study; the authors are willing to evaluate the utility of the Sepsis-3 criteria to diagnosis VAP, because chest X-ray lacks adequate sensitivity or specificity to diagnose pneumonia. However, according to the fact, the VAP diagnosis by chest X-ray in this study might not be accurate, and the diagnostic utility of Sepsis-3 criteria evaluated based on the inaccurate VAP diagnosis has no meaning. To prove the authors’ hypothesis, the gold-standard diagnosis of the VAP must be performed by more accurate modalities such as chest CT. The design and data set in this study cannot solve the question raised by the authors. It is necessary to comprehensively reconsider the study aim and design.

7. PLOS authors have the option to publish the peer review history of their article (what does this mean?). If published, this will include your full peer review and any attached files.

Reviewer #1: No

Reviewer #2: No

Reviewer #3: **Yes: **Kentaro Tojo

---

## [Author Response · Author response to Decision Letter 1]

19 Dec 2020

Dear Editor, 

Dear Reviewer, 

Thank you for allowing us to reply to your comments. 

We have provided some additional clarifications which, we believe, have improved the quality of our manuscript. 

Please find appended below our response to your concerns: 

Reviewer #3: Thank you very much for giving me the opportunity to review the revised version of the manuscript entitled “Usefulness of Sepsis-3 in diagnosing and predicting mortality of ventilator-associated lower respiratory tract infections”. I have understood the aim of the study; the authors are willing to evaluate the utility of the Sepsis-3 criteria to diagnosis VAP, because chest X-ray lacks adequate sensitivity or specificity to diagnose pneumonia. However, according to the fact, the VAP diagnosis by chest X-ray in this study might not be accurate, and the diagnostic utility of Sepsis-3 criteria evaluated based on the inaccurate VAP diagnosis has no meaning. To prove the authors’ hypothesis, the gold-standard diagnosis of the VAP must be performed by more accurate modalities such as chest CT. The design and data set in this study cannot solve the question raised by the authors. It is necessary to comprehensively reconsider the study aim and design.

We would like to thank you for this comment. As you are pointing out, our study evaluated the value of the Sepsis-3 criteria for the diagnosis of VAP, as the interpretation of the chest X-ray can be difficult in daily practice. 

Response: We fully understand your question regarding the relevance of the radiographic criteria in our study. Nevertheless, it should be remembered that the diagnosis of VAP was made with full knowledge of the patient's medical record. This included the possibility of observing the appearance of prior opacities on the chest X-ray, or secondarily unmasking a differential diagnosis of VAP, such as cardiac overload, atelectasis, or pleural effusion. In these different cases, the diagnosis of VAP could reasonably be made ultimately on the basis of the chest x-ray, while still being difficult in daily practice at the patient's bedside. Therefore, it seems to us that chest X-ray, which was used in the largest international study describing the characteristics of VAT and VAP , was relevant in our study.

On the other hand, CT scan was only used in a minority of cases to diagnose VAP in our patients, and therefore could not be used in our study. This situation reflects the low practical use of this tool to diagnose VAP. Thus, although we recognize the potential value of CT scan in a comprehensive and mechanistic study, its practical use for the diagnosis of VAP is likely to be limited. 

We are aware of the importance of these issues in the interpretation of our study. Accordingly, these different points have been developed in the limitations section of our manuscript. 

We hope that these explanations have helped to remove your concerns about the value of this study, and remain at your disposal to answer any further questions you may have.

---

## [Editor Report · Decision Letter 2]

2 Jan 2021

Usefulness of Sepsis-3 in diagnosing and predicting mortality of ventilator-associated lower respiratory tract infections

PONE-D-20-26296R2

Dear Dr. Gaudet,

We’re pleased to inform you that your manuscript has been judged scientifically suitable for publication and will be formally accepted for publication once it meets all outstanding technical requirements.

Kind regards,

Yutaka Kondo

Academic Editor

PLOS ONE

Additional Editor Comments (optional): Thank you for great effort to revise the manuscript. Current version is much improved and we decided this decision. Congratulations!
---

## [Editor Report · Acceptance letter]

6 Jan 2021

PONE-D-20-26296R2 

Usefulness of Sepsis-3 in diagnosing and predicting mortality of ventilator-associated lower respiratory tract infections 

Dear Dr. Gaudet:

I'm pleased to inform you that your manuscript has been deemed suitable for publication in PLOS ONE. Congratulations! Your manuscript is now with our production department. 

Kind regards, 

on behalf of

Dr. Yutaka Kondo 

Academic Editor

PLOS ONE